# The N-Oscillator Born–Kuhn Model: An In-Depth Analysis of Chiro-Optical Properties in Complex Chiral Systems

**DOI:** 10.3390/nano14030270

**Published:** 2024-01-26

**Authors:** Yiping Zhao, Andrei Galiautdinov, Jingzhi Tie

**Affiliations:** 1Department of Physics and Astronomy, The University of Georgia, Athens, GA 30602, USA; ag1@uga.edu; 2Department of Mathematics, The University of Georgia, Athens, GA 30602, USA; jtie@uga.edu

**Keywords:** Born–Kuhn model, chiral plasmonics, chiral structures, chiral optics, circular dichroism, optical rotatory dispersion, coupled oscillators

## Abstract

A comprehensive theory is developed for the chiral optical response of two configurations of the N-oscillator Born–Kuhn model (NOBK): the helically stacked and the corner stacked models. In the helical NOBK model, there is always a chiral response regardless of the value of *N*, whereas in the corner NOBK, only configurations with even *N* demonstrate a chiral response. Generally, the magnitudes of optical rotatory dispersion (ORD) and circular dichroism (CD) increase with *N* when the parameters of each oscillator are fixed. In cases of weak coupling, the spectral shapes of ORD and CD remain invariant, while strong coupling significantly alters the spectral shapes. For large damping, the spectral amplitude becomes smaller, and the spectral features become broader. In the presence of small damping, strong coupling introduces degeneracy in the coupled oscillator system, leading to multiple spectral features in both ORD and CD across the entire spectral region. This simple model can not only help in the design of tunable chiral metamaterials but also enhance our understanding of chiro-optical responses in structures with different configurations.

## 1. Introduction

Chiral metamaterials—the artificially created structures that use subwavelength building blocks to break the reflection symmetry—exhibit strong optical activity and possess chiral optical properties. The importance of chiral metamaterials lies in their ability to manipulate the polarization state of light and enable new functionalities that are not readily available in naturally occurring materials. By designing chiral metamaterials with tailored optical responses, researchers can develop novel devices such as circular polarizers [1], chiral lenses [2], and chiral absorbers [3] for applications in various fields, including optics, telecommunications, sensing [4], and imaging [5]. Understanding the chiral optical properties of these metamaterials is essential for harnessing their potential in practical applications. Chiral optical properties arise from the difference in the interaction of left-handed circularly polarized light (LCP) and right-handed circularly polarized light (RCP) with the metamaterial structures. When illuminated with circularly polarized light, chiral metamaterials induce a phase or absorption difference between LCP and RCP components, leading to a net optical activity characterized by optical rotatory dispersion (ORD) and circular dichroism (CD).

Numerical calculations are essential for understanding complex chiral structures and predicting their optical responses. Various methods, such as the finite-difference time-domain (FDTD) calculation [6], the finite element method (FM) [7], discrete dipole approximation (DDA) [8,9], and rigorous coupled-wave analysis (RCWA) [10], are commonly employed for such simulations. Numerical modeling allows researchers to optimize the design of chiral metamaterial structures for specific applications. However, interpreting the results and establishing a clear physical picture linking the calculated outcomes to the intricate structures can be challenging due to the complexity of the systems and numerical algorithms.

Recently, the Born–Kuhn (BK) model has emerged as a promising approach to address this challenge [11,12]. The BK model considers spatially stacked coupled oscillators with different orientations, providing a classic framework to predict the optical activity of chiral molecules. This model treats electrons in a molecule as damped Lorentzian oscillators subjected to an external electromagnetic wave, and it has been successfully applied to explain the chiral response of two perpendicular corner-stacked nanorods [13]. A generalized version of the BK model was later presented for stacked plasmonic nanorods with arbitrary azimuthal angles or polarization directions, enabling accurate predictions of optical and non-optical activity for various two-nanorod systems [14]. The BK model has also been extended to study chiroptical properties in systems with non-linear coupling, incorporating perturbative terms for each oscillator [15]. In a recent work, a systematic comparison between the BK model and the FDTD results has been conducted, demonstrating that these methods can successfully predict the chiro-optical properties of corner-stacked plasmonic nanorods [16].

However, in more complicated experimental scenarios, a greater number of oscillators has to be considered. For instance, Larsen et al. fabricated a triply stacked Ag oligomer with dielectric spacer layers in-between each of the Ag layers [17]. According to the correspondence principle from the BK model to plasmonic structures, each Ag layer can be treated as a damped harmonic oscillator, leading to a 3-oscillator BK model for the entire oligomer structure. In more complex scenarios, the number of plasmonic layers can be increased further, giving rise to an N-oscillator BK model. For example, in the Au-nanoparticle-decorated DNA structures, a chain of Au nanoparticles was arranged in a chiral manner around a central axis [18,19]. In this case, each Au nanoparticle acts as a damped harmonic oscillator, and the entire Au-nanoparticle-decorated DNA structure can be represented by an N-oscillator BK model. Moreover, Song et al. experimentally realized Au-nanoparticle-decorated double helical DNA structures [20], introducing additional complexity in configuration to the analogous BK model. Thus, it becomes important to theoretically extend the previously discussed 2-oscillator BK model to an N-oscillator BK model and investigate how the number of oscillators and the coupling among them influence the chiro-optical property. Experimental observations also indicate that the N-oscillator BK model may exhibit different oscillator configurations, which can significantly impact the resulting chiro-optical response. Therefore, it is expected that a systematic theoretical exploration of N-oscillator BK model holds great promise in providing valuable insights into the behavior of chiral structures with multiple oscillators and offers opportunities to elucidate the intricate mechanisms underlying their chiro-optical properties.

Here, we present a general theory for N-oscillator Born–Kuhn (NOBK) models with two different configurations: the helically stacked and corner stacked models. The exact analytic expressions for the ORD and CD responses in these models are derived, and their chiro-optical spectral features have been investigated systematically for different damping and coupling situations.

## 2. N-Oscillator Born–Kuhn Models

We consider two kinds of N-oscillator Born–Kuhn (NOBK) models, the helically stacked and corner stacked NOBK models as shown in Figure 1. As shown in Figure 1a, the helical NOBK model starts from an oscillator (blue, indicated by displacement *x*_1_) aligned with the positive *x*-axis. Moving up along with the *z*-axis, the second oscillator (red, displacement *y*_2_) is parallel to the positive *y*-axis, then the third oscillator (blue, displacement *x*_3_) is anti-parallel to the *x*-axis, the fourth oscillator (red, displacement *y*_4_) is aligned with the negative *y*-axis, and so on. Between the two adjacent oscillators, there is a coupling interaction, denoted by the dashed lines in the figure. Looking along the positive *z*-axis, all the oscillators are arranged to rotate counter-clockwise in Figure 1a. Clearly, a clockwise configuration can be designed as well. Both the clockwise helical NOBK and counter-clockwise helical NOBK models are mirror images of each other, and form a pair of enantiomers. For the corner stacked NOBK model shown in Figure 1b, the oscillators are arranged alternatively along the *x*- and *y*-axis, all the *x*-oscillators are aligned with the positive *x*-axis, while all *y*-oscillators are parallel to the *y*-axis. When the number of oscillators *N* is even, its mirror image is its enantiomer. However, if *N* is odd, it can be transformed into its mirror image by two rotations. For example, when *N* = 3, one can rotate the structure in Figure 1b 90^o^ clockwise around the *z*-axis, and then rotate it by 180^o^ about the *y*-axis to obtain its mirror arrangement.

Using the convention in which *positive* displacement always corresponds to particle’s motion in the positive direction of either x- or y-axis, and restricting consideration to identical damping coefficients γ, identical resontant frequencies $\omega_{0}^{2}$, as well as identical nearest-neighbors coupling constants g, the equations of motion for the oscillators in the helical NOBK model are
(1)$$\left\{\begin{array}{r} {\ddot{x}}_{1}+\gamma {\dot{x}}_{1}+\omega_{0}^{2}x_{1}+gy_{2}=f_{1}(t)\\ \begin{array}{c} {\ddot{y}}_{2}+\gamma {\dot{y}}_{2}+\omega_{0}^{2}y_{2}+gx_{1}-gx_{3}=f_{2}(t)\\ \begin{array}{c} {\ddot{x}}_{3}+\gamma {\dot{x}}_{3}+\omega_{0}^{2}x_{3}-gy_{2}+gy_{4}=f_{3}(t)\\ {\ddot{y}}_{4}+\gamma {\dot{y}}_{4}+\omega_{0}^{2}y_{4}+gx_{3}-gx_{5}=f_{4}(t)\\ \begin{array}{c} {\ddot{x}}_{5}+\gamma {\dot{x}}_{4}+\omega_{0}^{2}x_{5}-gy_{4}+gy_{6}=f_{5}(t)\\ \cdots \end{array} \end{array} \end{array} \end{array}\right.,$$
where the alternating ±g on the left-hand side are due to simple physical considerations, and there are *N* equations corresponding to *N* oscillators in the system. Since our primary interest is in the system’s gyrotropic response, we assume harmonic drive in the form of a plane electromagnetic wave of frequency ω and wave number k propagating in the positive z-direction,
(2)$$\left(\begin{array}{l} f_{1}(t)\\ f_{2}(t)\\ f_{3}(t)\\ f_{4}(t)\\ f_{5}(t)\\ \vdots \end{array}\right)=\frac{\left(-q_{e}\right)}{m_{e}}\left(\begin{array}{l} E_{x0}\\ E_{y0}e^{ikd}\\ E_{x0}e^{2ikd}\\ E_{y0}e^{3ikd}\\ E_{x0}e^{4ikd}\\ \vdots \end{array}\right)e^{-i(\omega t-kz_{0})},$$
where $q_{e}$ and $m_{e}$ are the effective charge and mass parameters characterizing the oscillating charge distributions. Applying the solutions,
(3)$$\left(\begin{array}{l} x_{1}(t)\\ y_{2}(t)\\ x_{3}(t)\\ y_{4}(t)\\ x_{5}(t)\\ \vdots \end{array}\right)=\left(\begin{array}{l} u_{1}\\ u_{2}\\ u_{3}\\ u_{4}\\ u_{5}\\ \vdots \end{array}\right)e^{-i(\omega t-kz_{0})},$$Equation (1) can be cast in a matrix form with steady-state solutions (i.e., all the $e^{-i\omega t}$ terms are dropped off),
(4)$$A_{N}\left(\begin{array}{l} u_{1}\\ u_{2}\\ u_{3}\\ u_{4}\\ u_{5}\\ \vdots \end{array}\right)=\frac{\left(-q_{e}\right)}{m_{e}}\left(\begin{array}{l} E_{x0}\\ E_{y0}e^{ikd}\\ E_{x0}e^{2ikd}\\ E_{y0}e^{3ikd}\\ E_{x0}e^{4ikd}\\ \vdots \end{array}\right),$$
with
(5a)$$A_{N}=\left(\begin{array}{llllll} \Omega^{2} & g & 0 & 0 & 0 & \cdots \\ g & \Omega^{2} & -g & 0 & 0 & \cdots \\ 0 & -g & \Omega^{2} & g & 0 & \cdots \\ 0 & 0 & g & \Omega^{2} & -g & \cdots \\ 0 & 0 & 0 & -g & \Omega^{2} & \cdots \\ \vdots & \vdots & \vdots & \vdots & \vdots & \ddots \end{array}\right),$$
where $\Omega^{2}\equiv \omega_{0}^{2}-\omega^{2}-i\gamma \omega .$ Note that the $A_{N}$ in Equation (5a) is for a counter-clockwise helical NOBK. For a clockwise helical NOBK, $A_{N}$ changes to
(5b)$$A_{N}=\left(\begin{array}{llllll} \Omega^{2} & -g & 0 & 0 & 0 & \cdots \\ -g & \Omega^{2} & g & 0 & 0 & \cdots \\ 0 & g & \Omega^{2} & -g & 0 & \cdots \\ 0 & 0 & -g & \Omega^{2} & g & \cdots \\ 0 & 0 & 0 & g & \Omega^{2} & \cdots \\ \vdots & \vdots & \vdots & \vdots & \vdots & \ddots \end{array}\right).$$In the following discussion for helical NOBK, we will focus on Equation (5a), the counter-clockwise helical NOBK.

For the corner NOBK model (Figure 1b), the equations of motion can be written as
(6)$$\left\{\begin{array}{r} {\ddot{x}}_{1}+\gamma {\dot{x}}_{1}+\omega_{0}^{2}x_{1}+gy_{2}=f_{1}(t)\\ \begin{array}{c} {\ddot{y}}_{2}+\gamma {\dot{y}}_{2}+\omega_{0}^{2}y_{2}+gx_{1}+gx_{3}=f_{2}(t)\\ \begin{array}{c} {\ddot{x}}_{3}+\gamma {\dot{x}}_{3}+\omega_{0}^{2}x_{3}+gy_{2}+gy_{4}=f_{3}(t)\\ {\ddot{y}}_{4}+\gamma {\dot{y}}_{4}+\omega_{0}^{2}y_{4}+gx_{3}+gx_{5}=f_{4}(t)\\ \begin{array}{c} {\ddot{x}}_{5}+\gamma {\dot{x}}_{4}+\omega_{0}^{2}x_{5}+gy_{4}+gy_{6}=f_{5}(t)\\ \cdots \end{array} \end{array} \end{array} \end{array}\right.,$$
and the steady-state solution for Equation (6) can also be written in matrix form with
(7a)$$A_{N}=\left(\begin{array}{llllll} \Omega^{2} & g & 0 & 0 & 0 & \cdots \\ g & \Omega^{2} & g & 0 & 0 & \cdots \\ 0 & g & \Omega^{2} & g & 0 & \cdots \\ 0 & 0 & g & \Omega^{2} & g & \cdots \\ 0 & 0 & 0 & g & \Omega^{2} & \cdots \\ \vdots & \vdots & \vdots & \vdots & \vdots & \ddots \end{array}\right).$$The $A_{N}$ for the mirror arrangement of Figure 1b about the *x–z* plane can be written as
(7b)$$A_{N}=\left(\begin{array}{llllll} \Omega^{2} & -g & 0 & 0 & 0 & \cdots \\ -g & \Omega^{2} & -g & 0 & 0 & \cdots \\ 0 & -g & \Omega^{2} & -g & 0 & \cdots \\ 0 & 0 & -g & \Omega^{2} & -g & \cdots \\ 0 & 0 & 0 & -g & \Omega^{2} & \cdots \\ \vdots & \vdots & \vdots & \vdots & \vdots & \ddots \end{array}\right).$$Similarly, below, we will only focus on Equation (7a), i.e., the structure in Figure 1b. The general solution for Equation (4) can be written as
(8)$$\left(\begin{array}{l} u_{1}\\ u_{2}\\ u_{3}\\ u_{4}\\ u_{5}\\ u_{6}\\ \vdots \end{array}\right)=-\frac{q_{e}}{m_{e}}A_{N}^{-1}\left(\begin{array}{l} E_{x0}\\ E_{y0}e^{ikd}\\ E_{x0}e^{2ikd}\\ E_{y0}e^{3ikd}\\ E_{x0}e^{4ikd}\\ E_{y0}e^{5ikd}\\ \vdots \end{array}\right).$$According to Appendix A, if $\phi_{l,j}$ is the element of the inverse matrix $A_{N}^{-1}$, then
(9)$$u_{l}=-\frac{q_{e}}{m_{e}}\sum_{j=1}^{N}\;\phi_{l,j}E_{j}e^{i\left(j-1\right)kd}=-\frac{q_{e}}{m_{e}}[\sum_{j=1}^{\left[\frac{N+1}{2}\right]}\phi_{l,\left(2j-1\right)}{e^{i2\left(j-1\right)kd}E}_{x0}+\sum_{j=1}^{\left[\frac{N}{2}\right]}\phi_{l,\left(2j\right)}{e^{i\left(2j-1\right)kd}E}_{y0}]={-\frac{q_{e}}{m_{e}}(u}_{lx}+u_{ly}).$$
where the symbol $\left[\frac{N+1}{2}\right]$ or $\left[\frac{N}{2}\right]$ means taking an integer less or equal to the term in [], $u_{lx}=\sum_{j=1}^{\left[\frac{N+1}{2}\right]}\phi_{l,\left(2j-1\right)}{e^{i2\left(j-1\right)kd}E}_{x0}$, and $\sum_{j=1}^{\left[\frac{N}{2}\right]}\phi_{l,\left(2j\right)}{e^{i\left(2j-1\right)kd}E}_{y0}$. For both the helical and corner NOBK models, the induced polarization can then be found as [21,22]
(10)$$P_{x}=\omega_{p}^{2}\sum_{l=0}^{\left[(N-1)/2\right]}u_{2l+1}{\left(\mp 1\right)}^{l}e^{-ik\left(2l\right)d}=\omega_{p}^{2}\sum_{l=0}^{\left[(N-1)/2\right]}{(u}_{(2l+1)x}{+u_{(2l+1)y})\left(\mp 1\right)}^{l}e^{-ik\left(2l\right)d},$$
(11)$$P_{y}=\omega_{p}^{2}\sum_{l=1}^{\left[N/2\right]}u_{2l}{\left(\mp 1\right)}^{l+1}e^{-ik\left(2l+1\right)d}=\omega_{p}^{2}\sum_{l=1}^{\left[N/2\right]}{(u}_{(2l)x}{+u_{(2l)y})\left(\mp 1\right)}^{l+1}e^{-ik\left(2l+1\right)d},$$
(12)$$P_{z}=0,$$
where $\omega_{p}^{2}=\frac{n_{0}q_{e}^{2}}{m_{e}}$, $n_{0}$ is the bulk concentration of NOBK molecules. The “−” sign is for the helical NOBK model, and the “+” sign is for the corner NOBK model. Insert Equation (9) into Equations (10) and (11), and one has
(13)$$P_{x}=\chi_{xx}E_{x0}+\chi_{xy}E_{y0}\;\mathrm{and}\;P_{y}=\chi_{yx}E_{x0}+\chi_{yy}E_{y0},$$
with
(14)$$\left\{\begin{array}{l} \chi_{xx}=\omega_{p}^{2}\sum_{l=0}^{\left[(N-1)/2\right]}\sum_{j=1}^{\left[(N+1)/2\right]}{\left(\mp 1\right)}^{l}\phi_{(2l+1),\left(2j-1\right)}e^{i\left(2j-2l-2\right)kd}\\ \chi_{xy}=\omega_{p}^{2}\sum_{l=0}^{\left[(N-1)/2\right]}\sum_{j=1}^{\left[N/2\right]}{{\left(\mp 1\right)}^{l}\phi }_{(2l+1),\left(2j\right)}e^{i\left(2j-2l-1\right)kd}\\ \begin{array}{l} \chi_{yx}=\omega_{p}^{2}\sum_{l=1}^{\left[N/2\right]}\sum_{j=1}^{\left[(N+1)/2\right]}{{\left(\mp 1\right)}^{l+1}\phi }_{(2l),\left(2j-1\right)}e^{i\left(2j-2l-3\right)kd}\\ \chi_{yy}=\omega_{p}^{2}\sum_{l=1}^{\left[N/2\right]}\sum_{j=1}^{N/2}{{\left(\mp 1\right)}^{l+1}\phi }_{(2l),\left(2j\right)}e^{i\left(2j-2l-2\right)kd} \end{array} \end{array}\right..$$

The exact expressions for polarization as a function of kd become rather cumbersome and not very illuminating for N>2, which motivates us to use the quasi-static (long-wavelength) approximation, kd≪1, and keep only linear terms in kd,
(15)$$\left\{\begin{array}{l} \chi_{xx}=\omega_{p}^{2}\sum_{l=0}^{\left[(N-1)/2\right]}\sum_{j=1}^{\left[(N+1)/2\right]}{\left(\mp 1\right)}^{l}\phi_{(2l+1),\left(2j-1\right)}\\ \chi_{xy}=\omega_{p}^{2}\sum_{l=0}^{\left[(N-1)/2\right]}\sum_{j=1}^{\left[N/2\right]}{{\left(\mp 1\right)}^{l}\phi }_{\left(2l+1\right),\left(2j\right)}[1+i\left(2j-2l-1\right)kd]\\ \begin{array}{l} \chi_{yx}=\omega_{p}^{2}\sum_{l=1}^{\left[N/2\right]}\sum_{j=1}^{\left[(N+1)/2\right]}{{\left(\mp 1\right)}^{l+1}\phi }_{(2l),\left(2j-1\right)}[1+i\left(2j-2l-3\right)kd]\\ \chi_{yy}=\omega_{p}^{2}\sum_{l=1}^{\left[N/2\right]}\sum_{j=1}^{\left[N/2\right]}{{\left(\mp 1\right)}^{l+1}\phi }_{(2l),\left(2j\right)} \end{array} \end{array}\right..$$Since $\phi_{l,j}=\phi_{j,l}$ for both helical and corner NOBK models, according to Equation (15), $\chi_{xy}$ and $\chi_{yx}$ can be written as [16,21,22],
(16)$$\chi_{xy}=\chi_{xy}^{0}-ik\Gamma \;\mathrm{and}\;\chi_{yx}=\chi_{yx}^{0}+ik\Gamma ,$$
with
(17)$$\chi_{xy}^{0}=\chi_{yx}^{0}=\omega_{p}^{2}kd\sum_{l=0}^{\left[(N-1)/2\right]}\sum_{j=1}^{\left[N/2\right]}{{\left(\mp 1\right)}^{l}\phi }_{\left(2l+1\right),\left(2j\right)},$$
(18)$$\Gamma =-\omega_{p}^{2}d\sum_{l=0}^{\left[(N-1)/2\right]}\sum_{j=1}^{\left[N/2\right]}{{\left(\mp 1\right)}^{l}\phi }_{\left(2l+1\right),\left(2j\right)}\left(2j-2l-1\right),$$
where Γ representing chiral induced polarization. If we only consider the chiral effect, after averaging in three-dimensional space, we obtain
(19)$$\Gamma =-\frac{\omega_{p}^{2}d}{3}\sum_{l=0}^{\left[(N-1)/2\right]}\sum_{j=1}^{\left[N/2\right]}{{\left(\mp 1\right)}^{l}\phi }_{\left(2l+1\right),\left(2j\right)}\left(2j-2l-1\right).$$

The chiral optical response of a chiral medium is determined by three optical parameters: the index of refraction, ORD, and CD. The index of refraction for RCP, denoted as $n_{+},$ and LCP, denoted as $n_{-},$ are distinct and can be represented as [16,23] $n_{\pm }^{2}={\bar{n}}^{2}\pm \kappa \bar{n}$ with $\bar{n}$ being the average index of refraction and κ is the gyration. In the limit of $\kappa \ll \bar{n}$, $\Delta n=n_{+}-n_{-}\approx \kappa$, which is determined by κ. According to [22], $\kappa \approx \frac{\omega }{v}\Gamma$ with v being the speed of the electromagnetic wave in vacuum. The ORD and CD can be calculated as follows,
(20a)$$ORD=\Delta \phi =\frac{\tilde{\omega }L}{2v_{l}}Re\left(\Delta n\right)\approx \frac{{\tilde{\omega }}^{2}}{2v_{l}^{2}}Re\left(\Gamma \right),$$
(20b)$$CD=\Delta A=\frac{2\tilde{\omega }L}{v_{l}}Im\left(\Delta n\right)\approx \frac{2{\tilde{\omega }}^{2}}{v_{l}^{2}}Im\left(\Gamma \right).$$
where Δϕ is optical rotation angle, ΔA is the change in the absorbance of the spectra due to the LCP and RCP incidence, $\tilde{\omega }=\frac{\omega }{\omega_{0}}$, $v_{l}=\frac{v}{\omega_{0}}$, and *L* is the path length of the medium. Since we can treat both $v_{l}$ and *L* as constant, we can define the effective ORD and CD responses,
(21a)$$O_{RD}=\frac{2v_{l}^{2}}{L}\times ORD={\tilde{\omega }}^{2}Re\left(\Gamma \right),$$
(21b)$$C_{D}=\frac{v_{l}^{2}}{2L}\times CD={\tilde{\omega }}^{2}Im\left(\Gamma \right).$$In the following section, we will discuss in detail how different NOBK models, the number of oscillators, and oscillator parameters affect the effective ORD and CD responses.

### 2.1. The Helical NOBK Model

For the helical NOBK model, according to Appendix A, $\phi_{l,j}$ can be written as
(22)$$\phi_{l,j}=(-1{)}^{\frac{j\left(j-1\right)-l\left(l-1\right)}{2}}\frac{\sin\left(l\sigma \right)\sin\left[\left(N+1-j\right)\sigma \right]}{gsin\sigma \sin\left[\left(N+1\right)\sigma \right]},$$
where $\sigma ={cos}^{-1}(\frac{\Omega^{2}}{2g})$. Thus,
(23)$$\Gamma =-\frac{\omega_{p}^{2}d}{3}\sum_{l=0}^{\left[(N-1)/2\right]}\sum_{j=1}^{\left[N/2\right]}(-1{)}^{j\left(2j-1\right)-l\left(2l+1\right)+l}\left(2j-2l-1\right)\frac{\sin\left((2l+1)\sigma \right)\sin\left[\left(N+1-2j\right)\sigma \right]}{gsin\sigma \sin\left[\left(N+1\right)\sigma \right]}.$$

Let us look at when *N* = 2, *l* = 0 and *j* = 2, so, $\Gamma =\frac{d\omega_{p}^{2}}{3}\frac{\sin\left[\sigma \right]}{g\sin\left[3\sigma \right]}=\frac{d\omega_{p}^{2}}{3g}\frac{1}{3-{4sin}^{2}\sigma }=\frac{d\omega_{p}^{2}}{3g}\frac{1}{4{\left(\frac{\Omega^{2}}{2g}\right)}^{2}-1}=\frac{d\omega_{p}^{2}}{3}\frac{g}{\Omega^{4}-g^{2}}$. Table 1 shows the calculated $\chi_{xx}$, $\chi_{xy}$, and Γ for some representative even *N* and odd *N* helical NOBKs respectively. We notice that when *N* is even, the expressions for $\chi_{xx}$, $\chi_{xy}$, and Γ become more complicated, while for odd *N* they are much simpler. Note that $\chi_{xy}=\chi_{yx}.$

Below, we will give an extensive discussion on how both ORD and CD change with *N* under different damping and coupling constants. To make all the quantities comparable, we set $b=\frac{\gamma }{\omega_{0}}$, and $c=\frac{g}{\omega_{0}^{2}}$, thus $\frac{\Omega^{2}}{\omega_{0}^{2}}=1-{\tilde{\omega }}^{2}-ib\tilde{\omega }$.

(1)Large damping

In cases where *b* is large and *c* is small, corresponding to weak coupling ($g\ll \Omega^{2})$, the expressions in Table 1 yield
(24)$$\Gamma \approx -\frac{(N-1)gd}{\Omega^{4}},$$
i.e., according to Equations (21a) and (21b), the functional shapes of both $O_{RD}$ and $C_{D}$ with respect to $\tilde{\omega }$ remain unchanged. Only the magnitudes of $O_{RD}$ and $C_{D}$ experience a linear increase with *N*. Figure 2a,b show the plots of $O_{RD}$ and $C_{D}$ versus $\tilde{\omega }$ for b=0.5 and c=0.001 at *N* = 2 to 9. The overall magnitudes of $\left|O_{RD}\right|$ and $\left|C_{D}\right|$ are smaller than 0.035. As expected, the overall amplitudes $\left|O_{RD}\right|$ and $\left|C_{D}\right|$ increase with *N*. $O_{RD}$ exhibits a primary peak and attains a maximum value at ${\tilde{\omega }}_{M}=1$. At ${\tilde{\omega }}_{z\mp }=0.78\;and\;1.28$, $O_{RD}$ reaches zero. These two zero positions are slightly asymmetry about ${\tilde{\omega }}_{M}=1$. Beyond these two ${\tilde{\omega }}_{z\mp }$ values, $O_{RD}$ is negative. Thus, for a fixed *c* with the increase in *N*, this $O_{RD}$ peak becomes sharper. On the contrary, $C_{D}$ exhibits a bisinuate line shape. At $\tilde{\omega }=1$, $C_{D}=0$. This observation is consistent across all values of *N*, *b*, or *c*, since at $\tilde{\omega }=1$, $\frac{\Omega^{2}}{\omega_{0}^{2}}=-ib\tilde{\omega }$ and according to the expressions for Γ in Table 1, the Γ value is real. Regardless of N values, at ${\tilde{\omega }}_{-}=0.866$, $C_{D}$ reaches a negative dip, while at ${\tilde{\omega }}_{+}=1.155$, $C_{D}$ achieve a positive peak. At these two extreme locations, $C_{D}\left({\tilde{\omega }}_{-}\right)=C_{D}\left({\tilde{\omega }}_{+}\right)$, while $1-{\tilde{\omega }}_{-}=0.134<{\tilde{\omega }}_{+}-1=0.155$, there is slight asymmetry in the $C_{D}$ spectrum about ${\tilde{\omega }}_{M}=1$. The slight asymmetric spectral shapes about ${\tilde{\omega }}_{M}=1$ in both $O_{RD}-\tilde{\omega }$ and $C_{D}-\tilde{\omega }$ spectra are due to strong damping.

In addition, $O_{RD}$ and $C_{D}$ are both influenced by the coupling strength *c*. Figure 2c,d present two-dimensional (2D) map plots of $O_{RD}-\tilde{\omega }$ and $C_{D}-\tilde{\omega }$ with varying *c* from 0.01 to 0.6 for *N* = 6. Several features can be seen: (1) The peak intensity of $O_{RD}-\tilde{\omega }$ consistently increases with *c*, as indicated by Equation (24), Γ∝c. (2) The $O_{RD}$ peak exhibits increased broadening with higher *c* values. (3) The separation of the negative dip location ${\tilde{\omega }}_{-}$ and positive peak location ${\tilde{\omega }}_{+}$ for $C_{D}$ increases with *c*. In more detail, Figure 3a shows plots of $O_{RD}$ and $C_{D}$ versus $\tilde{\omega }$ for b=0.5 and c=0.2 at *N* = 2 to 9. The overall spectral trends with *N* look similar to those of Figure 2a, while more detailed inspection shows some interesting differences. The maximum location ${\tilde{\omega }}_{M}$ of $O_{RD}-\tilde{\omega }$ spectra shifts slightly for different *N*: *N* = 2–5, ${\tilde{\omega }}_{M}=1.004$; when *N* increases to 6–9, ${\tilde{\omega }}_{M}=1.002.$ The zero-crossing locations ${\tilde{\omega }}_{z\pm }$ do not stay the same, rather they vary with *N*: the top of Figure 3c (red data points for *c* = 0.2) plots ${\tilde{\omega }}_{z\pm }-1$ versus *N*, and the two blue lines outline the ${\tilde{\omega }}_{z\pm }$ locations for *c* = 0.001. It appears that except for *N* = 2, with the increase in *N*, the ${\tilde{\omega }}_{z\pm }$ values at c=0.2 approach the corresponding values at c=0.001.

All the $C_{D}-\tilde{\omega }$ spectra maintain the bisinuate line-shape, but both ${\tilde{\omega }}_{-}$ and ${\tilde{\omega }}_{+}$ vary with *N*. The bottom of Figure 3c (red data points for *c* = 0.2) plots ${\tilde{\omega }}_{\pm }-1$ versus *N* with the two blue lines indicating the ${\tilde{\omega }}_{\pm }$ locations for *c* = 0.001. Similar trends like ${\tilde{\omega }}_{z\pm }$ versus *N* are observed for ${\tilde{\omega }}_{\pm }$. In addition, the $\left|C_{D}({\tilde{\omega }}_{-})\right|$ is always smaller than $\left|C_{D}({\tilde{\omega }}_{+})\right|$, further demostrating the asymmetric spectral shape. If we define a parameter $\gamma =\frac{C_{D}\left({\tilde{\omega }}_{+}\right)+C_{D}({\tilde{\omega }}_{-})}{\left|C_{D}({\tilde{\omega }}_{-})\right|+\left|C_{D}({\tilde{\omega }}_{+})\right|}$ to caharcetrize the degree of asymmetry, we obtain γ= 0.0339, 0.0658, 0.0458, 0.0339, 0.0279, 0.0224, 0.0193, and 0.0167 for *N* = 2 to 9, respectively. Except for *N* = 2, the γ value decreases monotonically with *N*, showing that the spectral shape of $C_{D}-\tilde{\omega }$ becomes less asymetric.

As the coupling strength *c* increases to 0.6, indicating a scenario of strong damping and strong coupling, both $O_{RD}-\tilde{\omega }$ and $C_{D}-\tilde{\omega }$ spectra for *N* = 2 to 9 closely resemble those at *c* = 0.2, with distinct ${\tilde{\omega }}_{M}$, ${\tilde{\omega }}_{z\pm }$, and ${\tilde{\omega }}_{\pm }$ for a fixed *N*, as shown in Figure 3b. For *N* = 2, 4, 6, and 8, both the $O_{RD}-\tilde{\omega }$ and $C_{D}-\tilde{\omega }$ spectra are quite similar to those shown in Figure 3a. Detailed inpsction shows that ${\tilde{\omega }}_{M}$ shifts significantly away from ${\tilde{\omega }}_{M}=1$: ${\tilde{\omega }}_{M}=$ 1.078, 1.031, 1.019, and 1.014 for *N* = 2, 4, 6, and 8, respectively, i.e., ${\tilde{\omega }}_{M}$ decreases monotonically with even number *N*. The $O_{RD}$ spectral shapes become more asymetric compared to those of *c* = 0.2. However, for odd *N*, the spectral shapes are highly skewed. For *N* = 3. The peak at $\tilde{\omega }=1$ is very broad with $O_{RD}$ value at $\tilde{\omega }<1$ significantly smaller than that at $\tilde{\omega }>1$. With the increase in *N*, the peak becomes narrower, but still skewed, with left-side $O_{RD}$ values smaller than right-side $O_{RD}$ values. Only when *N* = 9 does the $O_{RD}-\tilde{\omega }$ spectrum become more symmetric. In addition, for *N* = 3, 5, 7, and 9, ${\tilde{\omega }}_{M}=$ 1.166, 1.078, 1.038, and 1.021, also decreases monotonically with odd number *N*. Similar behaviors are observed for $C_{D}$ spectra: for even *N*s, the change in $C_{D}$ spectra versus *N* seems continuing the trend from Figure 3a, while for odd *N*s, the spectra become highly asymmetric.

The black data points in Figure 3c (top figure) show the relationship between ${\tilde{\omega }}_{z\pm }-1$ and *N* for *c* = 0.6. Both ${\tilde{\omega }}_{z\pm }$ are quite far away from the ${\tilde{\omega }}_{z\pm }$ locations at *c* = 0.001. Except for *N* = 2, ${\tilde{\omega }}_{z+}-1$ approaches 0.28 (the *c* = 0.001 value) monotonically with the increase in *N*; similar relationship is observed for ${\tilde{\omega }}_{+}-1$ in the bottom of Figure 3c. For ${\tilde{\omega }}_{z-}$, no monotonical trend is observed. However, for ${\tilde{\omega }}_{-}-1$, depending on whether *N* is even or odd, it seems to follow two monotonically increasing trends as indicated by the dotted and dashed purple curves (guides for eyes). The degree of asymmetry of the $C_{D}$ spectrum is also characterized by γ, with γ= 0.253, 0.178, 0.175, and 0.149 for *N* = 2, 4, 6, and 8, respectively; as well as γ= 0.475, 0.252, 0.162, and 0.113 for *N* = 3, 5, 7, and 9. Therefore, the γ value decreases monotonically with even or odd *N*. This shows that with the increase in *N*, the chiro-optical response spectra will become more symmetric. Note that for *c* = 0.6, all the γ values are almost one order of magnitude large than those at *c* = 0.2, indicating that stronger coupling between adjacent oscillators will induce more asymmetric chiro-optical response.

(2)Small damping

Figure 4 plots the selected $O_{RD}-\tilde{\omega }$ and $C_{D}-\tilde{\omega }$ relationships for *N* = 2 to 9 at *b* = 0.01 and *c* = 0.001, 0.2, and 0.6, respectively. In the scenario where c=0.001≪b=0.01, corresponding to small damping and weak coupling, both $O_{RD}-\tilde{\omega }$ and $C_{D}-\tilde{\omega }$ have the same spectral shape as shown in Figure 4a. The $O_{RD}-\tilde{\omega }$ spectra are symmetric about $\tilde{\omega }=1$, with ${\tilde{\omega }}_{M}=1.00$ for all *N*, and only two fixed zero-crossings locations are observed, with ${\tilde{\omega }}_{z-}=$ 0.995 and ${\tilde{\omega }}_{z+}=$1.005, both symmetrically located about ${\tilde{\omega }}_{M}=1$. In addition, the $O_{RD}$ spectra are notably sharper than those in Figure 2a and the corresponding maximum values ($O_{RD}(\tilde{\omega }=1))$ are in the order of 10–80, significantly larger than those in situations with larger damping. For $C_{D}-\tilde{\omega }$ spectra, only 1 pair of ${\tilde{\omega }}_{-}$ and ${\tilde{\omega }}_{+}$ are observed at ${\tilde{\omega }}_{-}=$ 0.997 and ${\tilde{\omega }}_{+}=$ 1.003, regardless of *N*, also symmetrically located about ${\tilde{\omega }}_{M}=1$. Furthermore, we observe that $\left|C_{D}({\tilde{\omega }}_{-})\right|=\left|C_{D}({\tilde{\omega }}_{+})\right|$. The small separation between ${\tilde{\omega }}_{-}$ and ${\tilde{\omega }}_{+}$ results in a small spectral span in $C_{D}$ but much greater magnitudes compared to Figure 2b. The magnitude of both $O_{RD}$ and $C_{D}$ increases monotonically with *N*.

When *c* increases to 0.2, the spectral features in both $O_{RD}-\tilde{\omega }$ and $C_{D}-\tilde{\omega }$ relationships become more complicated as shown in Figure 4b. For *N* = 3, 5, 7, and 9, the $O_{RD}-\tilde{\omega }$ spectra share a similar shape, featuring a positive curved band centered around $\tilde{\omega }=1$ and two large negative dips near each zero-crossing location, with the width of the central band decreasing with *N*. For *N* = 2, 4, 6, and 8, the central bands are relatively narrower than their *N* + 1 counterparts. While the spectrum is asymmetry about $\tilde{\omega }=1$, there are $\left[N/2\right]$ zero crossings in $\tilde{\omega }>1$ and $\tilde{\omega }<1$ regions, respectively. Near each zero crossing, there is either a negative dip or a positive peak. Similar features are observed for the $C_{D}-\tilde{\omega }$ relationship: For *N* = 3, 5, 7, and 9, there is only a negative dip at $\tilde{\omega }<1$ and a positive peak at $\tilde{\omega }>1$, with specific (${\tilde{\omega }}_{-}$, ${\tilde{\omega }}_{+}$) pairs being (0.845, 1.135), (0.895, 1.095), (0.920, 1.075), and (0.935, 1.060), respectively. For *N* = 2, 4, 6, and 8, $\left[N/2\right]$ positive peaks and *N*/2 negative dips are evident in the spectra. Specifically, for *N* = 2, ${\tilde{\omega }}_{-}$ and ${\tilde{\omega }}_{+}$ appear at 0.895 and 1.095; for *N* = 4, two negative dips occur at 0.935 and 0.82, and two positive peaks at 1.06 and 1.15; for *N* = 6, there are three negative dips at 0.865, 0.955, 1.165, and three positive peaks at 1.12, 1.045, 0.80; for *N* = 8, four dips are at 0.965, 0.895, 0.79, and 1.145, and four peaks at 1.035, 1.095, 1.175, and 0.835. Clearly, the number of peaks/dips in the $C_{D}$ plots corresponds to the number of zero crossings in the $O_{RD}$ plots.

As *c* increases to 0.6, signifying a strong coupling case, the $O_{RD}-\tilde{\omega }$ and $C_{D}-\tilde{\omega }$ relationships become even more complicated as shown in Figure 4c. All the $O_{RD}$ and $C_{D}$ spectra become markedly asymmetric, with espectral features stretched in $\tilde{\omega }<1$ region. For *N* = 2, 3, 4, 5, 6, 7, and 8, both the $O_{RD}$ and $C_{D}$ spectra closely resemble those at *c* = 0.2, yet with increased asymmetry and expanded separations of relative peaks or dips. However, for *N* = 9, two zero-crossing locations emerge for $O_{RD}$ at $\tilde{\omega }<1$ and for $C_{D}$, negative dips appear at 0.79 and 0.17, accompanied by two positive peaks at 1.17 and 1.4.

To better understand the observed splitting behaviors for different values of *c*, Figure 5 presents the 2D maps of $C_{D}-\tilde{\omega }$ as *c* varies from 0.001 to 0.6. The darker line-like features in the plots represent sharp dips while the bright curves represent sharp peaks. Initially, all maps exhibit one dip and one peak at very small *c*, and depending on *N* and *c*, these dip and peak split into multiple dip and peak lines. For *N* = 2, 3, 5, and 7, the dominate features in the $C_{D}$ maps consist of one dip line and one peak line at $\tilde{\omega }<1$ and $\tilde{\omega }>1$, respectively. With the incease in *c*, the separation between ${\tilde{\omega }}_{-}$ and ${\tilde{\omega }}_{+}$ increases monotonically. However, for the same *c*, the ${\tilde{\omega }}_{-}$ and ${\tilde{\omega }}_{+}$ seperations become smaller with increased *N*. In contrast, for even *N* (>2), multiple dips and peaks emerge. For example, for *N* = 4, two distinct dip lines appear at $\tilde{\omega }<1$ and two peak lines occur at $\tilde{\omega }>1$. These four lines remain in both *N* = 6 and *N* = 8 maps. But in the *N* = 6 map, a faint peak line emerges to the left of the second dip line, and a very weak dip line appears to the right of the second peak line. For *N* = 8, compared to *N* = 6, a very weak dip line is added on the far left, while a weak peak line is introduced on the far right.

The splitting of the peaks/dips for large *c* arises from the degeneracy of the coupled oscillators in a weak damping case. When *c* is very small, the dominant oscillation modes in the NOBK system are the bonding and anti-bonding modes. As discussed in [16], the ${\tilde{\omega }}_{-}$ and ${\tilde{\omega }}_{+}$ correspond to bonding and anti-bonding modes of the system for *N* = 2. In fact, as *c* increases, the stronger coupling between two adjacent oscillators leads to the degeneration of oscillation modes. Since *b* is very small, the NOBK system can be treated as *N*-coupled harmonic oscillators with an intrinsic frequency of $\omega_{0}$. Due to the coupling, the new collective oscillation mode $\omega_{k}$ becomes [24],
(25)$${\tilde{\omega }}_{k}=\frac{\omega_{k}^{2}}{\omega_{0}^{2}}=1-2ccos(\frac{k\pi }{N+1}),\;k=1,2,\cdots ,N.$$

By numerically examining the negative dip and positive peak locations of $C_{D}$, we find that these locations are exactly corresponding to all the collected modes ${\tilde{\omega }}_{k}$ for *N* = 2*m* and some selected modes for *N* = 2*m* + 1, where *m* = 1, 2, 3,… In fact, an equation ${\tilde{\omega }}_{k}=\sqrt{1-2a_{k}c}$ can be used to fit all these $\tilde{\omega }$ locations, with $a_{k}=cos\left(\frac{k\pi }{N+1}\right),\;k=1,\;2,\;\ldots ,m$. For *N* = 2*m*, according to Equation (25), there are a total of 2*m* resonant modes emerging, which correspond to the *m* dips and *m* peaks shown in the top row of Figure 5. By fitting these locations, we find that each of the collective modes of the *N* = 2*m* BK oscillators can exhibit a chiral response. For *N* = 2*m* + 1, when *k* = *m* + 1, ${\tilde{\omega }}_{m+1}=1$, meaning that each oscillator in the NOBK model vibrates with its own intrinsic frequency, results in the absence of a chiral response. Therefore, for *N* = 3, only two chiral modes with ${\tilde{\omega }}_{1}=\sqrt{1+\sqrt{2}c}$ and ${\tilde{\omega }}_{3}=\sqrt{1-\sqrt{2}c}$ are present. For *N* = 5, except for the ${\tilde{\omega }}_{m+1}=1$, only when the modes of ${\tilde{\omega }}_{2}=\sqrt{1+c}$ and ${\tilde{\omega }}_{4}=\sqrt{1-c}$ demonstrate chiral response, which is consistent for when *N* = 2. For *N* = 7, we found four *a* values, $a_{1,7}=\pm cos\left(\frac{\pi }{8}\right)$ and $a_{3,5}=\pm cos\left(\frac{3\pi }{8}\right)$, which correspond to ${\tilde{\omega }}_{1}$ and ${\tilde{\omega }}_{7}$, ${\tilde{\omega }}_{3}$ and ${\tilde{\omega }}_{5}$ modes, with chiral responses. For *N* = 9, also only four modes have active chiral responses, $a_{2,8}=\pm cos\left(\frac{2\pi }{10}\right)$ and $a_{4,6}=\pm cos\left(\frac{4\pi }{10}\right)$, due to ${\tilde{\omega }}_{2}$ and ${\tilde{\omega }}_{8}$, ${\tilde{\omega }}_{4}$ and ${\tilde{\omega }}_{6}$ modes. Thus, for *N* = 2*m* + 1, all modes ${\tilde{\omega }}_{m+1\pm 2j}$ with $j\leq \left[\frac{m}{2}\right]$ do not exhibit chiral response. Such a result is due to the intrinsic mirror semmetry of the collective osscilation of these modes. Let us take *N* = 5 for example, the eigen vectors for ${\tilde{\omega }}_{1}-{\tilde{\omega }}_{5}$ are $\left\{1,\;\sqrt{3},-2,-\sqrt{3},\;1\right\}$, $\left\{-1,-1,\;0,-1,\;1\right\}$, $\left\{1,\;0,\;1,\;0,\;1\right\}$, $\left\{-1,\;1,\;0,\;1,\;1\right\}$, and $\left\{1,-\sqrt{3},-2,\;\sqrt{3},\;1\right\}$, respectively. The clearly eigen vector for ${\tilde{\omega }}_{3}$ shows non-zero *x*-component oscillators, and they are on the same plane, while the eigen vectors for ${\tilde{\omega }}_{1}$ and ${\tilde{\omega }}_{5}$ are mirror vectors about the *x–z* plane: the amplitudes for the first *y*-oscillator and second *y*-oscillator are interchangeable. Therefore, these two modes do not show chiral response. Similar features are observed for the eigen vectors for ${\tilde{\omega }}_{2}$ and ${\tilde{\omega }}_{6}$ for when *N* = 7, and eigen vectors for ${\tilde{\omega }}_{1}$ and ${\tilde{\omega }}_{9}$, ${\tilde{\omega }}_{3}$ and ${\tilde{\omega }}_{5}$ for when *N* = 9.

### 2.2. The Corner NOBK Model

For the corner NOBK model, according to Section A.2, the element of the inverse matrix $A_{N}$ (Equation (7)) can be written as
(26)$$\phi_{l,j}=(-1{)}^{l+j}\frac{sin(l\sigma )sin((N+1-j)\sigma )}{gsin\sigma sin((N+1)\sigma )},$$

Thus,
(27)$$\Gamma =\frac{\omega_{p}^{2}d}{3}\sum_{l=0}^{\left[(N-1)/2\right]}\sum_{j=1}^{\left[N/2\right]}\left(2j-2l-1\right)\frac{sin((2l+1)\sigma )sin((N+1-2j)\sigma )}{gsin\sigma sin((N+1)\sigma )}.$$

Due to the structural symmetry of the model in Figure 1b, when *N* is odd, Γ=0. Therefore, only when *N* is even, the corner NOBK model has a non-vanishing Γ, i.e., has a chiro-optical response. The resulting expressions of $\chi_{xx}$, $\chi_{xy}$, and Γ for some even *N* are summarized in Table 2. Here we also have $\chi_{xy}=\chi_{yx}$.

Figure 6 plots some $O_{RD}-\tilde{\omega }$ and $C_{D}-\tilde{\omega }$ spectra for different *b* and *c*. When c≪b, $\Gamma_{2}=\Gamma_{4}\approx -\frac{gd}{\Omega^{4}}$ and $\Gamma_{6}=\Gamma_{8}\approx -\frac{2gd}{\Omega^{4}}$, i.e., the spectral shape of the chiral response for *N* = 2 and *N* = 4 are the same, same for *N* = 6 and *N* = 8. Figure 6a,c show the $O_{RD}-\tilde{\omega }$ and $C_{D}-\tilde{\omega }$ spectra for b=0.5 (large damping) and c=0.01 and b=0.01 (small damping) with the same c=0.001. For both cases, due to the generancy, only two curves exist. In fact, on the basis of the above discussion, though the amplitudes of $O_{RD}-\tilde{\omega }$ and $C_{D}-\tilde{\omega }$ spectra for *N* = 2 and 4 are distinct, the rescaled spectral shapes should be exactly the same, which means both the ${\tilde{\omega }}_{z\pm }$ and ${\tilde{\omega }}_{\pm }$ are the same for fixed *b* and *c*. However, compared when b=0.5, the spectra for when b=0.01 are much narrower, more symmetric, and have a much greater magnitude in both $O_{RD}$ and $C_{D}$. However, when *c* increases, all spectra start to degenerate. For example, as shown in Figure 6b, for b=0.5 and c=0.2, four different spectra emerge for both $O_{RD}$ and $C_{D}$. Similar trend is observed for b=0.01 and c=0.2 as shown in Figure 6d. The behavior of the spectra at high damping constnat (say b=0.5) are very similar to those discussed in the chiral NOBK model (Figure 2 and Figure 3). However, for small damping, the behavior is very different. The spectral shapes of $O_{RD}-\tilde{\omega }$ mimic multiple-band structure. For *N* = 2, there is a positive band between $\tilde{\omega }=0.894$ and 1.096. For *N* = 4, two positive bands appear, one is between $\tilde{\omega }=0.823$ and 0.936, the other between $\tilde{\omega }=1.06$ and 1.151. For *N* = 6, three positive bands are shown, $\tilde{\omega }=0.8$ to 0.866, 0.955 to 1.043, and 1.118 to 1.167. In between these positive bands, there are two negative bands. Through Figure 6d, it is interesting to note that the positive bands in $O_{RD}$ spectra for different *N* are largely compensate to different spectral regions with slightly overlaps at the edge. It is expected that by combining helical NOBK layers with different *N* and appropriate thickness, one may design a broader band optical rotator.

For $C_{D}-\tilde{\omega }$, there are multiple peaks and dips quite symmetrically distributed around $\tilde{\omega }$ = 1, each spectrum has *N*/2 numbers of peaks and equal number of dips. In fact, both the adjacent peaks and dips can be treated as a bisinuate line shape, with different zero-cross locations and corresponding ${\tilde{\omega }}_{\pm }$, i.e., there are *N*/2 numbers of bisinuate lines. For *N* = 2, there is only one bisinuate line shape, with zero-crossing location at ${\tilde{\omega }}_{z}$ = 1 and ${\tilde{\omega }}_{\pm }=$1.096 and 0.895. For *N* = 4, the first zero-crossing location ${\tilde{\omega }}_{z1}$ = 1.106 and corresponding ${\tilde{\omega }}_{1\pm }=$1.151 and 1.06; the second zero-crossing at ${\tilde{\omega }}_{z2}$ = 0.882 and corresponding ${\tilde{\omega }}_{2\pm }=$0.936 and 0.823. For *N* = 6, there are three bisinuate line shapes, with ${\tilde{\omega }}_{z1}$ = 1.141, ${\tilde{\omega }}_{1\pm }=$1.167 and 1.118; ${\tilde{\omega }}_{z2}$ = 1, ${\tilde{\omega }}_{2\pm }=$1.044 and 0.955; and ${\tilde{\omega }}_{z3}$ = 0.836, ${\tilde{\omega }}_{3\pm }=$0.867 and 0.8. Finally for *N* = 8, the four bisinuate line shapes are ${\tilde{\omega }}_{z1}$ = 1.158, ${\tilde{\omega }}_{1\pm }=$1.173 and 1.143; ${\tilde{\omega }}_{z2}$ = 1.066, ${\tilde{\omega }}_{2\pm }=$1.096 and 1.034; ${\tilde{\omega }}_{z3}$ = 0.931, ${\tilde{\omega }}_{3\pm }=$0.965 and 0.895; and ${\tilde{\omega }}_{z4}$ = 0.812, ${\tilde{\omega }}_{4\pm }=$0.833 and 0.79. If one take ${\tilde{\omega }}_{m}$ = 1 as the reference location and inspect the $C_{D}$ responses at $\tilde{\omega }<1$ and $\tilde{\omega }>1,$ one can find that the responses of *N* = 2 and 6 are exactly out of phase of response from *N* = 4 and 8. Thus, $C_{D}$ responses of even *N*/2 models are opposite to the response of odd N/2 models. Compared to the weak coupling case, the magnitudes of both $O_{RD}$ and $C_{D}$ are approximately one order of magnitude larger.

## 3. Conclusions

In conclusion, our work presents a comprehensive theory elucidating the chiral optical response of two distinct *N*-oscillator Born–Kohn models: the helically stacked and the corner stacked configurations. Each model comprises *N* identical damped oscillators with uniform coupling strength between adjacent oscillators. Our findings reveal that in the helical NOBK model, a chiral response is consistently observed irrespective of the value of *N*, which is due to the intrinsic mirror symmetry-breaking arrangement. However, the corner NOBK model exhibits chiral response only in configurations with even *N*, as odd *N* oscillators possess mirror symmetry and do not exhibit chiral response. Furthermore, our study demonstrates that the magnitudes of ORD and CD monotonically increase with *N* when the parameters of each oscillator are held constant. Specifically, we consider two scenarios: large damping and small damping. In instances of weak coupling, the spectral shapes of ORD and CD remain unchanged, whereas strong coupling induces significant alterations in their spectral shapes. For large damping, both ORD and CD exhibit small spectral amplitudes, with relatively simple and broad spectral features. In contrast, for small damping, strong coupling introduces degeneracy in the coupled oscillator system, resulting in multiple zero crossings in the ORD spectrum and multiple peaks/dips in the CD spectrum. The number of zero-crossings for ORD and peaks/dips for CD is directly related to *N*. In particular for CD, the collective eigen vibrations of an even number of helical oscillators correspond to the dips and peaks observed, while for an odd number of helical oscillators, only those collective modes with no mirror symmetry eigen vector pairs can demonstrate chiral response.

This comprehensive theoretical framework can not only enhance our understanding of chiro-optical responses in structures with similar configurations but also help us design specific chiral metamaterials. The model can be used to explain the chiral response of helically and corner stacked chain molecules, such as DNA or DNA like structures [25]. It can also be used to explain the chiro-optical response of chiral metamaterials. For example, for the helically Au-nanoparticle-decorated DNA structures shown in Figure 7a [18,19], since each Au NP can be treated as a damped plasmonic oscillator [13,16], the number of Au NPs decorated on the DNA will be determined by the length of the DNA. It is expected from our model that one can change the length of the DNA to change the number *N* of the Au NPs, so that the chiro-optical response of the structure can be tuned. Similarly, if one could tune the diameter of the Au NP, one can effectively tune the damping and coupling between adjacent Au NPs (due to the change in the gap *d* between adjacent Au NPs, as shown in Figure 7a), and, therefore, tune the chiro-optical response. In addition, the theoretical model can be used to design new chiral metamaterials to meet specific requirements. For helically stacked plasmonic materials realized by Larsen et al. [17] or helically stacked plasmonic bars shown in Figure 7b, the damping parameter *b* of the Ag layers is experimentally determined by the thickness and quality of the deposited plasmonic materials and the intrinsic material properties [26,27,28]. If the quality of the deposited material is poor, the damping *b* will be large, one can observe broad and simple chiral responses like those shown in Figure 2, regardless of the coupling strength *c*. However, if high-quality material is used and deposited, resulting in a small *b*, then the chiral optical response of the system can be tuned by the coupling strengths as well as the number of plasmonic layers. The coupling between adjacent oscillators is determined by the thickness and dielectric property of the thin dielectric layer between two plasmonic layers shown in Figure 7b. By appropriately choosing the material and the thickness of the insulating layers (such as SiO_2_ in [17]), one can decrease or increase the coupling strength *c*. Another design parameter is the number of plasmonic layers, *N*. Therefore, once the parameters *b* and *c* can be experimentally adjusted to relate to deposition conditions and structural configurations, one can implement the theoretical equations obtained here to help in designing desired chiral metamaterials with specific chiro-optical response.

## Figures and Tables

**Figure 1 nanomaterials-14-00270-f001:**
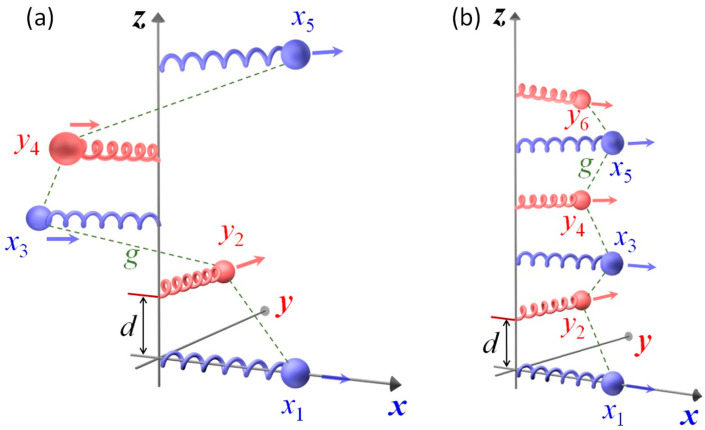
Schematic representation of (**a**) helically stacked and (**b**) corner stacked N-oscillator Born–Kuhn models. In the illustration, all the blue oscillators are either parallel or anti-parallel to the *x*-axis, while all red oscillators are either parallel or anti-parallel to the *y*-axis.

**Figure 2 nanomaterials-14-00270-f002:**
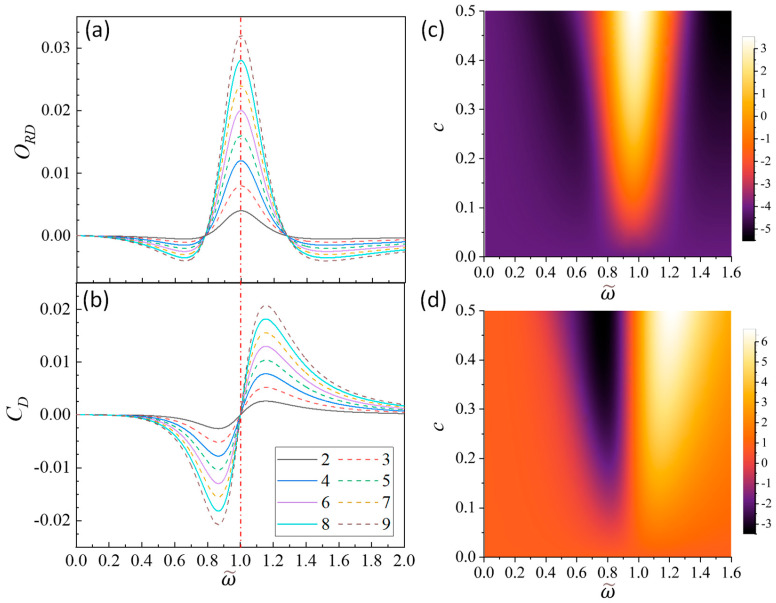
The plots of (**a**) $O_{RD}-\tilde{\omega }$ and (**b**) $C_{D}-\tilde{\omega }$ for b=0.5 and c=0.001 at *N* = 2 to 9. The 2D maps of (**c**) $O_{RD}$ and (**d**) $C_{D}$ with 0.01≤c≤0.6 for *N* = 6.

**Figure 3 nanomaterials-14-00270-f003:**
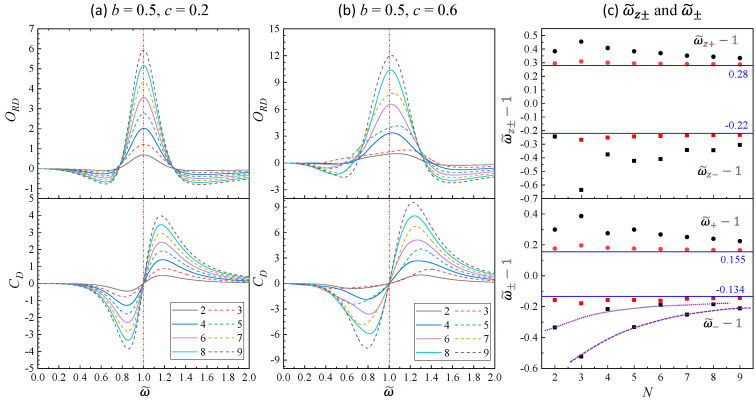
The plots of $O_{RD}-\tilde{\omega }$ and $C_{D}-\tilde{\omega }$ at *N* = 2 to 9 for b=0.5 and (**a**) c=0.2 and (**b**) c=0.6. (**c**) The plots of ${\tilde{\omega }}_{z\pm }-1$ and ${\tilde{\omega }}_{\mp }-1$ versus *N* for c = 0.2 (red) and 0.6 (black), respectively.

**Figure 4 nanomaterials-14-00270-f004:**
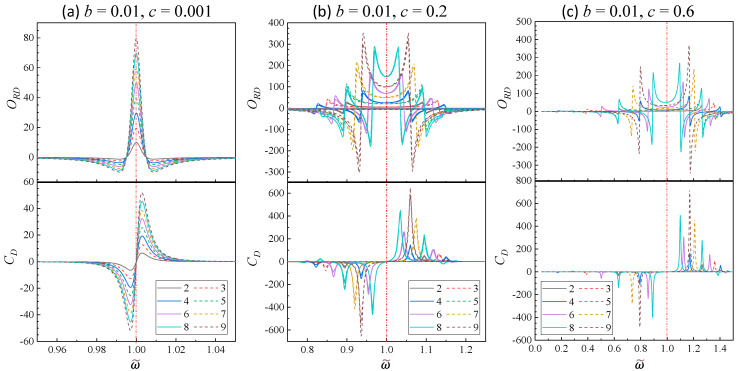
The plots of $O_{RD}-\tilde{\omega }$ and $C_{D}-\tilde{\omega }$ at *N* = 2 to 9 for b=0.01 and (**a**) $c=0.001,\;\left(b\right)\;c=0.2$, and (**c**) c=0.6.

**Figure 5 nanomaterials-14-00270-f005:**
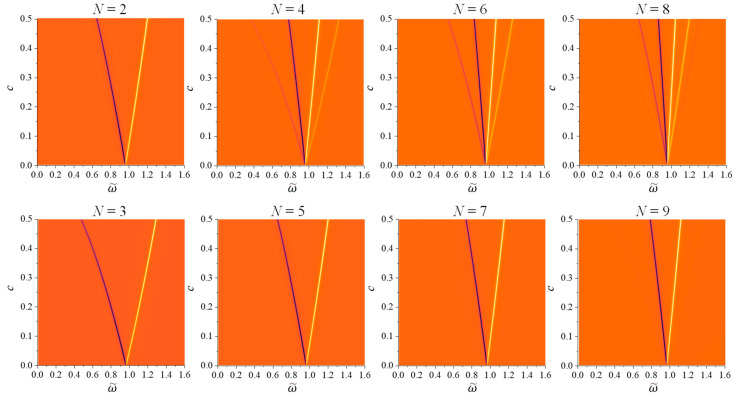
The 2D maps of $C_{D}-\tilde{\omega }$ for c=0.001 to 0.6 at different *N*.

**Figure 6 nanomaterials-14-00270-f006:**
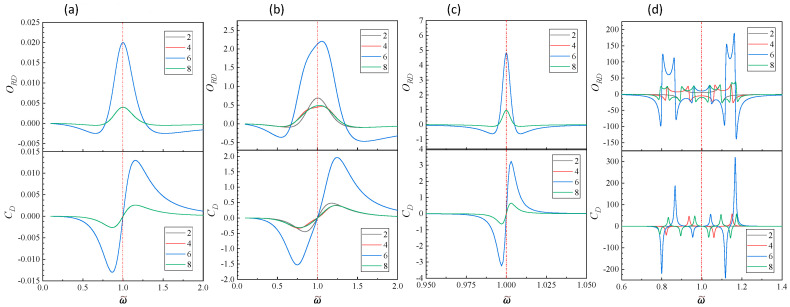
The plots of $O_{RD}-\tilde{\omega }$ and $C_{D}-\tilde{\omega }$ of the corner NOBK models at *N* = 2, 4, 6, and 8 for (**a**) b=0.5 and $c=0.001;\;\left(b\right)\;b=0.5\;and\;c=0.2$; (**c**) b=0.01 and c=0.001; and (**d**) b=0.01 and c=0.2.

**Figure 7 nanomaterials-14-00270-f007:**
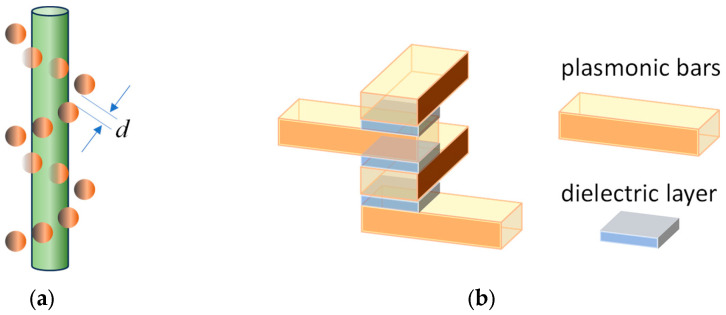
(**a**) Helically decorated Au NPs and (**b**) helically stacked plasmonic bars with a thin dielectric layer.

**Table 1 nanomaterials-14-00270-t001:** The resulting $\chi_{xx},\;\chi_{yy}$, $\chi_{xy}$, and Γ expression for helical NOBK models.

N	$\chi_{xx}=\chi_{yy}$	$\chi_{xy}$	Γ
2	$\frac{-\Omega^{2}}{g^{2}-\Omega^{4}}$	$\frac{g}{g^{2}-\Omega^{4}}$	$\frac{gd}{g^{2}-\Omega^{4}}$
4	$\frac{\Omega^{2}\left(-5g^{2}+2\Omega^{4}\right)}{g^{4}-3g^{2}\Omega^{4}+\Omega^{8}}$	$\frac{3g^{3}-g\Omega^{4}}{g^{4}-3g^{2}\Omega^{4}+\Omega^{8}}$	$\frac{gd\left(5g^{2}-3\Omega^{4}\right)}{g^{4}-3g^{2}\Omega^{4}+\Omega^{8}}$
6	$\frac{\Omega^{2}\left(-14g^{4}+14g^{2}\Omega^{4}-3\Omega^{8}\right)}{g^{6}-6g^{4}\Omega^{4}+5g^{2}\Omega^{8}-\Omega^{12}}$	$\frac{g\left(6g^{4}-5g^{2}\Omega^{4}+\Omega^{8}\right)}{g^{6}-6g^{4}\Omega^{4}+5g^{2}\Omega^{8}-\Omega^{12}}$	$\frac{gd\left(14g^{4}-21g^{2}\Omega^{4}+5\Omega^{8}\right)}{g^{6}-6g^{4}\Omega^{4}+5g^{2}\Omega^{8}-\Omega^{12}}$
8	$\frac{\Omega^{2}\left(-30g^{6}+54g^{4}\Omega^{4}-27g^{2}\Omega^{8}+4\Omega^{12}\right)}{g^{8}-10g^{6}\Omega^{4}+15g^{4}\Omega^{8}-7g^{2}\Omega^{12}+\Omega^{16}}$	$\frac{g\left(2g^{2}-\Omega^{4}\right)\left(5g^{4}-5g^{2}\Omega^{4}+\Omega^{8}\right)}{g^{8}-10g^{6}\Omega^{4}+15g^{4}\Omega^{8}-7g^{2}\Omega^{12}+\Omega^{16}}$	$\frac{gd\left(30g^{6}-81g^{4}\Omega^{4}+45g^{2}\Omega^{8}-7\Omega^{12}\right)}{g^{8}-10g^{6}\Omega^{4}+15g^{4}\Omega^{8}-7g^{2}\Omega^{12}+\Omega^{16}}$
10	$\frac{\Omega^{2}\left(-55g^{8}+154g^{6}\Omega^{4}-132g^{4}\Omega^{8}+44g^{2}\Omega^{12}-5\Omega^{16}\right)}{g^{10}-15g^{8}\Omega^{4}+35g^{6}\Omega^{8}-28g^{4}\Omega^{12}+9g^{2}\Omega^{16}-\Omega^{20}}$	$\frac{g\left(15g^{8}-35g^{6}\Omega^{4}+28g^{4}\Omega^{8}-9g^{2}\Omega^{12}+\Omega^{16}\right)}{g^{10}-15g^{8}\Omega^{4}+35g^{6}\Omega^{8}-28g^{4}\Omega^{12}+9g^{2}\Omega^{16}-\Omega^{20}}$	$\frac{gd\left(55g^{8}-231g^{6}\Omega^{4}+220g^{4}\Omega^{8}-77g^{2}\Omega^{12}+9\Omega^{16}\right)}{g^{10}-15g^{8}\Omega^{4}+35g^{6}\Omega^{8}-28g^{4}\Omega^{12}+9g^{2}\Omega^{16}-\Omega^{20}}$
3	$\frac{2}{\Omega^{2}}$	$\frac{-\Omega^{2}}{2g^{2}-\Omega^{4}}$	$\frac{2gd}{2g^{2}-\Omega^{4}}$
5	$\frac{3}{\Omega^{2}}$	$\frac{-2\Omega^{2}}{g^{2}-\Omega^{4}}$	$\frac{4gd}{g^{2}-\Omega^{4}}$
7	$\frac{4}{\Omega^{2}}$	$\frac{\Omega^{2}\left(3\Omega^{4}-10g^{2}\right)}{2g^{4}-4g^{2}\Omega^{4}+\Omega^{8}}$	$\frac{2gd\left(10g^{2}-3\Omega^{4}\right)}{2g^{4}-4g^{2}\Omega^{4}+\Omega^{8}}$
9	$\frac{5}{\Omega^{2}}$	$\frac{2\Omega^{2}\left(2\Omega^{4}-5g^{2}\right)}{g^{4}-3g^{2}\Omega^{4}+\Omega^{8}}$	$\frac{4gd\left(5g^{2}-2\Omega^{4}\right)}{g^{4}-3g^{2}\Omega^{4}+\Omega^{8}}$

**Table 2 nanomaterials-14-00270-t002:** The resulting $\chi_{xx},\;\chi_{yy}$, $\chi_{xy}$, and Γ expression for corner NOBK models.

N	$\chi_{xx}=\chi_{yy}$	$\chi_{xy}$	Γ
2	$\frac{-\Omega^{2}}{g^{2}-\Omega^{4}}$	$\frac{g}{g^{2}-\Omega^{4}}$	$\frac{gd}{g^{2}-\Omega^{4}}$
4	$\frac{\Omega^{2}\left(2\Omega^{4}-g^{2}\right)}{g^{4}-3g^{2}\Omega^{4}+\Omega^{8}}$	$\frac{g^{3}-3g\Omega^{4}}{g^{4}-3g^{2}\Omega^{4}+\Omega^{8}}$	$-\frac{dg\left(g^{2}+\Omega^{4}\right)}{g^{4}-3g^{2}\Omega^{4}+\Omega^{8}}$
6	$\frac{\Omega^{2}\left(2g^{4}-6g^{2}\Omega^{4}+3\Omega^{8}\right)}{-g^{6}+6g^{4}\Omega^{4}-5g^{2}\Omega^{8}+\Omega^{12}}$	$\frac{2g^{5}-9g^{3}\Omega^{4}+5g\Omega^{8}}{g^{6}-6g^{4}\Omega^{4}+5g^{2}\Omega^{8}-\Omega^{12}}$	$\frac{dg\left(2g^{4}-g^{2}\Omega^{4}+\Omega^{8}\right)}{g^{6}-6g^{4}\Omega^{4}+5g^{2}\Omega^{8}-\Omega^{12}}$
8	$\frac{\Omega^{2}\left(-2g^{6}+14g^{4}\Omega^{4}-15g^{2}\Omega^{8}+4\Omega^{12}\right)}{g^{8}-10g^{6}\Omega^{4}+15g^{4}\Omega^{8}-7g^{2}\Omega^{12}+\Omega^{16}}$	$\frac{2g^{7}-21g^{5}\Omega^{4}+25g^{3}\Omega^{8}-7g\Omega^{12}}{g^{8}-10g^{6}\Omega^{4}+15g^{4}\Omega^{8}-7g^{2}\Omega^{12}+\Omega^{16}}$	$-\frac{dg\left(2g^{6}+3g^{4}\Omega^{4}-3g^{2}\Omega^{8}+\Omega^{12}\right)}{g^{8}-10g^{6}\Omega^{4}+15g^{4}\Omega^{8}-7g^{2}\Omega^{12}+\Omega^{16}}$
10	$\frac{\Omega^{2}\left(3g^{8}-26g^{6}\Omega^{4}+48g^{4}\Omega^{8}-28g^{2}\Omega^{12}+5\Omega^{16}\right)}{-g^{10}+15g^{8}\Omega^{4}-35g^{6}\Omega^{8}+28g^{4}\Omega^{12}-9g^{2}\Omega^{16}+\Omega^{20}}$	$\frac{3g^{9}-39g^{7}\Omega^{4}+80g^{5}\Omega^{8}-49g^{3}\Omega^{12}+9g\Omega^{16}}{g^{10}-15g^{8}\Omega^{4}+35g^{6}\Omega^{8}-28g^{4}\Omega^{12}+9g^{2}\Omega^{16}-\Omega^{20}}$	$\frac{dg\left(3g^{8}-3g^{6}\Omega^{4}+8g^{4}\Omega^{8}-5g^{2}\Omega^{12}+\Omega^{16}\right)}{g^{10}-15g^{8}\Omega^{4}+35g^{6}\Omega^{8}-28g^{4}\Omega^{12}+9g^{2}\Omega^{16}-\Omega^{20}}$

## Data Availability

All data are presented in the manuscript.

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
