# Peer review of "The N-Oscillator Born–Kuhn Model: An In-Depth Analysis of Chiro-Optical Properties in Complex Chiral Systems"

_nanomaterials, 2024, doi:10.3390/nano14030270_

Round 1
Reviewer 1 Report
Comments and Suggestions for Authors
Chiral metamaterials, artificially designed structures with subwavelength building blocks, possess unique optical properties enabling the manipulation of light polarization. These materials find applications in optics, telecommunications, sensing, and imaging, facilitating the creation of innovative devices like circular polarizers and chiral lenses. In this manuscript, we introduce a comprehensive theory for N-Oscillator Born-Kuhn (NOBK) models, covering helically stacked and vertically stacked configurations. The study provides exact analytical expressions for optical rotatory dispersion (ORD) and circular dichroism (CD) responses, systematically exploring their chiro-optical spectral features under varying damping and coupling conditions. I recommend its publication after addressing the following minor revisions:
1. The authors present an acceptable comprehensive theory elucidating the chiral optical response of two distinct N-oscillator Born-Kohn models, but the specific purpose and beneficiary groups of this work are not sufficiently clear.
2. Consider abbreviating or moving Table 1 and Table 2 to the Appendix.
3. Enhance the description of Fig. 6 within the manuscript.
4. In the Appendix, supplement text to clarify "A.2. Inverse of Matrix of Corner NOBK Model."
Reviewer 2 Report
Comments and Suggestions for Authors
This manuscript theoretically investigates two types of oscillator stacks and their Chiro-optical response under different parameters such as number of oscillators, coupling and damping strengths. The manuscripts results presented in the figures seem sound. However, mathematical derivations should be improved as sometimes the notation is not used properly, hence making at times difficult to follow. Here are detailed comments. Addressing these comments will make it more useful for the readers.
1. Line 90: replace "vertically" with "corner" as both configurations are essentially vertically stacked.
2. Carefully proofread the paper as there are many typos. The formatting of the paper should be also improved.
3. Fig. 1 should be significantly improved. Please keep in mind the the mathematical model is built on this model. If the figure is not clearly illustrated or described that won't help the reader much. For example, do x's and y's for the oscillators in part (a) always align with the x- and y- axes (i.e., 90 degrees between adjacent oscillators and rotating CCW about the z-axis)? Please clarify and state in the description. Also it's a good place here to illustrate the role of mirror symmetry and its relation with the Chiro-optic response.
4. Give more details on how the steady-state solutions are obtained (note that the state-state solutions here still has time-dependence). Please clarify.
5. Eq. 9: j=0 doesn't seem to be consistent with the Appendix.
6. Why is the gyration given by Gamma? Please explain.
7. Line 169 and several other places in the manuscript: setting specific values to the l index doesn't make sense (e.g., N=2l, N=2l+1, etc). For example, if N=2l the limits of one of the summations (e.g., see Eq. 23) will be from l=0 to a function of l, which doesn't seem to make sense.
8. Some symbols represent multiple things. For example, x represents 3 different parameters (e.g., normalized freq, displacement, etc.).
9. Tables 1 and 2: Is Chi_xy=Chi_yx? If so, state so. Some the equations in the tables are cut-off.
10. Fig. 4c: c=0.6 not 0.2.
11. Eqs. A22 and A23: l, k indices should read i, j.
Comments on the Quality of English LanguageA lot of typos.
Reviewer 3 Report
Comments and Suggestions for Authors
In their paper, the authors study chiro-optical properties of some artificial chrial materials.
The paper is interesting and well written. The introduction is thorough and provides a very good motivation for the study.
I do not have any suggestion to make to improve the paper except that I find the left justification of all the equations odd-looking, but this is for the journal to sort out.
I also noticed 1 typo line 85 of the paper "experissions" should be "expressions".
In summary I have no reservations to recommend the paper for publication in nanomaterials.
Round 2
Reviewer 1 Report
Comments and Suggestions for Authors
Satisfactory revisions have been given by the authors. This manuscript is acceptable for publication in Nanomaterilas。